# VAD Patients’ Perception of Potential Telemedicine Support

**DOI:** 10.3390/ijerph19073768

**Published:** 2022-03-22

**Authors:** Thomas Berg, Carina Benstoem, Ajay K. Moza

**Affiliations:** 1Department of Thoracic and Cardiovascular Surgery, Medical Faculty RWTH Aachen, 52074 Aachen, Germany; amoza@ukaachen.de; 2Department of Operative Intensive Care Medicine and Intermediate Care, Medical Faculty RWTH Aachen, 52074 Aachen, Germany; cbenstoem@ukaachen.de

**Keywords:** VAD patients, patient support, telemedicine, acceptance, quality of care

## Abstract

Technical possibilities for patient support must be user-friendly. This includes acceptance on the part of the patients, and safe function that must not lead to the user being overtaxed. In a study of Ventricular Assist Device (VAD) patients at the Department of Thoracic and Cardiovascular Surgery at the University Hospital RWTH Aachen, a questionnaire was used to investigate the current situation in dealing with the VAD system. This was followed by a query about ideas, wishes, and fears regarding the increased use of telemedical applications. An adapted Service User Technology Acceptability Questionnaire (SUTAQ) was used and the evaluation was carried out with the help of Office-based applications. As a result, it can be stated that the patients do not want to see personal contacts and care be completely replaced by telemedical remote support. If the application is stable and functioning, the majority is very much open to such support.

## 1. Introduction

The use of ventricular assist devices (VAD) has become a cornerstone in the treatment of terminal heart failure, as figures from 2019 clearly indicate in Germany, 333 heart transplants were performed during this period, while 973 VAD systems were implanted [1]. The survival times on the VAD system are increasingly similar to the survival times after transplantation of a donor organ. VAD patients are dependent on a 24/7 easily accessible specialist care for a lifetime. This need arises from frequent potential complications with the system that can occur in VAD patients over time, e.g., bleeding complications, increased risk of stroke, driveline infections, or defects in the device or accessories.

We intend to gain information for managerial and practical concerns. Early detection of developments that can lead to life threatening situations is mandatory. The use of telemedical access—here in the form of telemonitoring [2]—to a VAD center can alleviate the need for on-site examination of the patient. Such remote monitoring is currently not the standard of follow-up care for VAD patients while at home. Although the VADs are technically already able to access the web in individual cases, the existing interface technology has not yet reached market maturity [3].

The current COVID-19 pandemic strongly advocates the use of remote telemonitoring, not only in VAD patients [4]. In current times, prescient patient management unequivocally mandates the expansion of telemedical areas of application and capacities. To date, great efforts are being made to meet these ends [5]. 

At present, patients and VAD specialists alike are making strenuous efforts to ensure adequate monitoring of VAD systems. Telemedicine is a powerful tool to reduce the burden on the health care system (elimination of the journey, fewer waiting times, etc.). Furthermore, the reduction of hospital admissions as a result of prescient detection and treatment of complications would also be a result of such remote monitoring. All sides would benefit from telemedical applications; time and costs could be noticeably reduced, and approaches and ideas to improve Quality of Life (QoL) and lifetime with VAD support can be seen here.

From November 2018 to January 2019, we conducted a survey among patients who have been supported with a VAD system for at least one year. Fifty patients took part and provided information about personal data and their current situations in everyday life with a VAD system. 

Furthermore, they commented on a number of questions that capture expectations for the use of a telemedicine system and from which conclusions can be drawn about specific requirements for a telemedical application. We aimed to assess how remote monitoring can be established in VAD patients and how acceptance of telemedicine is propagated in this cohort. A self-developed questionnaire based on the Service User Technology Acceptability Questionnaire (SUTAQ) was used to address these ends.

The results of the study should then support the implementation of a telemedical application according to the needs of patients and the technical requirements.

## 2. Materials and Methods

### 2.1. Study Design

In this retrospective study, we anonymously asked VAD patients (November 2018–January 2019) during their regular outpatient follow up about their opinion on additional telemedical care after VAD implantation. Only those VAD patients who had been on the system for at least one year were included in this study. The patients were randomly selected during the VAD consultation.

### 2.2. Ethical Approval 

Ethical Approval for the Study was Granted by the Local Ethics Committee (EK 108/15) 

### 2.3. Description of the Questionnaire

For this purpose, we adapted the Service User Technology Acceptability Questionnaire (SUTAQ) to our needs. The current situation with the VAD system and basic demographic data were the subject of the survey.

The SUTAQ was developed for the Whole Systems Demonstrator Study (WSD) in the UK. The aim was to measure the acceptance of technical health services and to identify characteristics of people who are likely to reject their use. The SUTAQ contains 22 points that indicate on a Likert scale of 1 to 6 how far the respondents agree with the statements [6].

We interviewed these patients about positive aspects of, as well as possible hurdles to, introducing additional telemedical care.

The questionnaire was divided into three parts, in which a total of 37 questions had to be answered.

Part 1 of the questionnaire dealt with general personal information. Gender, age, height, and weight were recorded, followed by marital status, domestic situation, support at home, professional situation. Finally, questions about the state of health and physical resilience followed. For this purpose, 10 questions were asked; the possible answers varied. 

Part 2 of the survey focuses on the current situation of the patient and their daily challenges with the VAD system. Here, the in-hospital days in the current year and in the previous year, the number of visits to the outpatient department (OPD) (unscheduled and in total in the last 12 months), and the frequency of telephone contacts with the OPD in the last 12 months were of interest. This information was supplemented with data regarding the distance from the patients’ residence to the clinic, duration and circumstances of the journey, and the estimated waiting time in the OPD, as well as information on patients’ satisfaction with the care provided by the VAD team. Besides the frequency of visits to the OPD, the frequency of data retrieval from the VAD and its documentation, as well as the method and frequency of INR controls at home were assessed. Finally, the patients indicated at what intervals they would suggest that the data from the VAD should be retrieved from the VAD team in the OPD. For this purpose, a total of 16 questions were asked, in which numbers, checkboxes, and free text inputs were combined.

In the third part, patients were asked about their ideas on when to start using telemedical remote monitoring. Expectations and fears when using telemedical applications were documented. In a total of 21 questions, agreement or disagreement with standardized statements were collected using Likert scales of 1 to 6 (6 meaning strong agreement; 1 meaning strong disagreement). In addition, there were possibilities for free text input (twice) and checkboxes (twice). The answers based on the Likert scale were then aggregated to be able to express the agreement or disagreement in percentages.

### 2.4. Statistical Analysis

The analysis was carried out in such a way that the questionnaires were transferred manually to Excel-based documents; the analysis of the collected data was carried out by means of descriptive statistics. Here, the results are presented with the help of common Microsoft Office (Redmond, WA, USA) pie charts.

The information from the free text fields was categorized after entry; hence, the corresponding categories were created after the material was examined, (inductive approach) [7].

## 3. Results

### 3.1. Population Characteristics

The evaluation of the first part of the questionnaire showed that the participating patients were on average between 45 and 80 years old at the time of the survey (on average 65 years); the age distribution showed that 4% (*n* = 2) were 50 years or younger, 24% (*n* = 12) were between 51 and 60 years, 42% (*n* = 21) were between 61 and 70 years, and 30% (*n* = 15) were 70 years and older (Figure 1a). 

Of the patients, 90% (*n* = 45) were men and 10% were women (*n* = 5) (Figure 1b), while 70% (*n* = 35) lived in a partnership (Figure 1c).

Of the patients, 68% (*n* = 34) rated their state of health as okay to very good (Figure 2a), which was expressed in terms of the physical resilience in 48% (*n* = 24) of the respondents with a walking distance of more than 200 meters in six minutes (Figure 2b).

It turned out that 78% (*n* = 39) of the VAD patients were able to access support at home (Figure 3b). In 66% (*n* = 33) of the cases, family members and/or friends were the supporters, 24% (*n* = 12) had professional help, and in 10% (*n* = 5) of the cases, patients were either on their own or dependent on sporadic supporters (Figure 3c) (Multiple answers were possible in answering this question.)

The questionnaire revealed that most patients with a VAD system were no longer able to pursue regular employment—96% (*n* = 48) of the patients were without permanent employment and receiving pension or social welfare benefits (Figure 3a).

### 3.2. Device-Related Clinical Contacts

In the second part, the survey showed large differences among patients with regard to the number of their in-hospital stays in the year 2017 and 2018; 22% (*n* = 11) in 2017 and 30% (*n* = 15) in 2018 were not hospitalized at all; 30% (*n* = 15) in 2017 and 44% (*n* = 22) in 2018 were hospitalized for up to 15 days; 44% (*n* = 22) (2017) and 22% (2018) had 16 days or more in the clinic. The longest stay recorded was 124 days. On average, the patients spent 19.48 days (2017) and 11.42 days (2018) as inpatients in hospitals.

Number of visits to the outpatient clinic were recorded with a mean of 6.81 days in 2018, (Table 1) with an average distance from the place of residence to the clinic of about 50 km (maximum 260 km) and an average travel time of longer than 30 min in 52% (*n* = 26) of the cases. The same percentage of patients 52% (*n* = 26) see at least partial or comprehensive problems with the accessibility of the clinic (Table 2), which fits into the picture.

The patients assessed the retrieval of the VAD data as well as their transmission frequency and review by the clinic as follows. A daily control of VAD parameters was carried out by 62% (*n* = 31) by the patients themselves; 32% (*n* = 16) did this twice daily, while 6% (*n* = 3) left greater intervals between controls. The frequency of transmission to the clinic (currently within the framework of routine appointments at intervals of three months) appeared for 82% (*n* = 41) to be sufficient in the current setting, while 6% (*n* = 3) wanted shorter intervals.

### 3.3. Acceptance and Concerns about a Telemedical Connection

The third part of the questionnaire, which aims to reveal acceptance and concerns regarding a telemedical access, is divided into six areas. Here, various statements were provided and patients’ agreement/disagreement recorded. These questions were formulated in such a way to assess the patients’ take on following topics:(1)Improvement of the health situation or the state of care;(2)Easier access to help;(3)Privacy and discomfort;(4)Patients’ concerns about the staff involved;(5)Telemedicine use as a complement to face-to-face contact;(6)Potential satisfaction with telemedicine application.

These six individual areas were examined in sub-groups focusing on different aspects. For this purpose, the respondents rated given statements on a scale of 1 to 6 from strong agreement (6) to strong disagreement (1). Additional remarks could be made in individual areas in free text forms. 

#### 3.3.1. Improvement of the Health Situation/the Condition of Care

The first part of the query "Improvement of the health situation/the condition of care" was split into five sub-groups. The first subgroup dealt with tangible improvement of the health situation. Here, 42% (*n* = 21) of respondents had positive expectations associated with the use of telemedicine. Equally, 42% (*n* = 21) expressed uncertainty about this, and 16% (*n* = 8) assumed an adverse development (Figure 4a).

This was followed by a query of expectations to be better able to report experiences back to the clinic while on VAD support in order to improve support for oneself and other VAD patients. Of the patients, 48% (*n* = 24) of the patients expect a faster exchange and 70% (*n* = 35) also expect that this will result in a positive development not only for themselves.

In total, 56% (*n* = 28) expected an improvement in care, 34% (*n* = 17) were uncertain, and only 10% (*n* = 5) saw little / no chance that their own ideas of improved care could be put in practice (Figure 4b).

For VAD patients, improvement of accessibility and logistics related to life on a VAD system was an issue (24%, *n* = 12), as well as better care and realization of treatment options (22%, *n* = 11). Timely exchange of information and improved training with the VAD as a result of a telemedical connection (14%, *n* = 7) seemed to be equally important. Finally, future visionary medical solutions (e.g., the possibility of better artificial organs in the future) were formulated by 4% (*n* = 2). 

#### 3.3.2. Easier Access Help

The second main topic dealt with "Easier access to help". The first question was if the patient assume that they would need to spend less time to obtain professional help, and if the support needed would be easier to obtain. Of respondents, 50% (*n* = 25) expected that it would be faster to contact the VAD team and that support would be easier to obtain (60%, *n* = 30).

There was skepticism regarding the impact on decreasing the number of clinic contacts (inpatient and outpatient); 42% (*n* = 21) did not expect less inpatient contact and 52% (*n* = 26) did not expect less outpatient appointments, while 14% (*n* = 7) made no statement.

#### 3.3.3. Privacy

Regarding privacy and concerns in this context, it can be said that few problems are feared both with regard to the security of one’s own data (68%, *n* = 34) (Figure 5a) and with regard to the practical aspect of telemedical applications (58%, *n* = 29) (Figure 5b). Accordingly, the practical application is conceivable for the participants without the negative effects of the described burdensome factors.

#### 3.3.4. Concerns about the Quality of Care

Patient concerns about the expertise of staff involved in the remote monitoring and the quality of care delivered were raised by 38% (*n* = 19), while 60% (*n* = 30) saw no negative potential here (Figure 6a). With regard to the deployment of staff in the context of telemedicine, 58% (*n* = 29) of the respondents did not expect any negative effects (Figure 6b).

#### 3.3.5. Telemedicine as a Supplement

Telemedical application as a supplement to personal contact was examined under three aspects. Only 10% (*n* = 5) expected negative effects on their health status, whereas 78% (*n* = 39) did not expect such adverse effects by telemedical support (Figure 7a).

Of patients, 56% (*n* = 28) do not expect fewer checkups in outpatient clinic; however 30% (*n* = 15) can imagine that this could be achieved in the future (Figure 7b); 68% (*n* = 34) of respondents do not share the expectation that personal contacts can be completely replaced, while 20% (*n* = 10) disagree (Figure 7c). 

#### 3.3.6. Potential Satisfaction with a Telemedical Application

Of the respondents, 54% (*n* = 27) can imagine an equally good standard of care with telemedicine compared to the clinic, while 44% (*n* = 22) expressed skepticism/rejection.

The use of telemedicine as add-on was supported by 32% (*n* = 16), of which 20% (*n* = 10) see reliable functionality of the technical telemedical equipment as a precondition; only 4% (*n* = 2) would reject any telemedical remote monitoring at all. 

A total of 42% (*n* = 21) expect an improvement of day to day care through telemedicine, 30% (*n* = 15) are undecided, and 24% (*n* = 12) do not believe such an improvement will occur.

## 4. Discussion

Telemedical remote management surely cannot replace current in-hospital or outpatient (56%) and personal contacts (76%). Telemedical support as add on to the existing level of care is positively seen by the vast majority of patients without significant concerns about data security (76%). There are also no concerns about the quality of care when using a combination of familiar outpatient care and telemedical support (66%). If the technical requirements for a stable and reliable telemedical support are met, patients would prefer the additional support to gain more independence and mobility. Given the lack of a nationwide telemedical infrastructure, it is of no surprise that patients cling to face-to-face examinations. However, the current pandemic mandates the development of such infrastructure with reliable and user-friendly telemedical devices, not only for VAD patients. Telemedicine is a powerful tool to prevent current healthcare systems from collapsing. Avoiding needless hospital admissions while preventing catastrophic complications with prescient remote patient management seems more than warranted these days [8,9]. 

On the provider side, this requires an adjustment of structures and, ultimately, an expansion of resources to ensure a reliable telemedicine network [2]. As clinicians, it is our task to implement telemedical remote management in hospitals without compromising on the quality of care we seek to deliver.

As we have shown, our VAD patients would surely appreciate the implementation of such technology. The instant exchange of clinically relevant information with the clinic and the increase in independence and mobility seem to be tempting. As with every innovative technology, it will take time for our patients to become acquainted with it. Certain patient groups are already embracing this new technology, provided that support and user-friendliness are given [10]. Eventually, the benefits for the health care system as well as for the patients themselves will clearly reward efforts to implement telemedicine in daily clinical practice. 

## 5. Conclusions

Patients expect safe and manageable support that takes their needs into account and adds value without complications. Replacing face-to-face contact is not a desirable option for the majority. If access to support can be made easier and journeys and other logistical burdens can be reduced, this is an effect that the majority would welcome. Against the current pandemic background, these expectations are even more pressing and deserve priority attention.

We were able to show that the majority of respondents, regardless of age group, are fundamentally open to developments in telemedicine structures. Developing applications with concrete benefits and good usability and making them accessible should be the goal of increased efforts in the coming years.

## 6. Limitations

Not all patients who are eligible for VAD therapy in our clinic were explicitly asked, and the reasons why some patients decline VAD therapy were documented. It was also not recorded whether the presence or absence of telemedical support could be the reason for this.

We are aware that *n* = 50 is too small a sample size to generate statistically significant differences. However, it may help to see a trend in favor of telemedical support.

## Figures and Tables

**Figure 1 ijerph-19-03768-f001:**
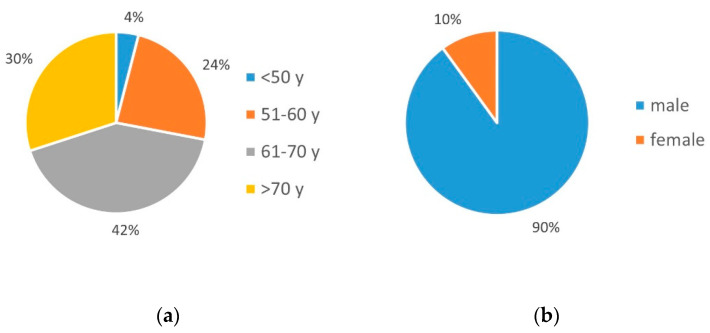
The characteristics of the study group: (**a**) age (Y-years), (**b**) sex, (**c**) marital status.

**Figure 2 ijerph-19-03768-f002:**
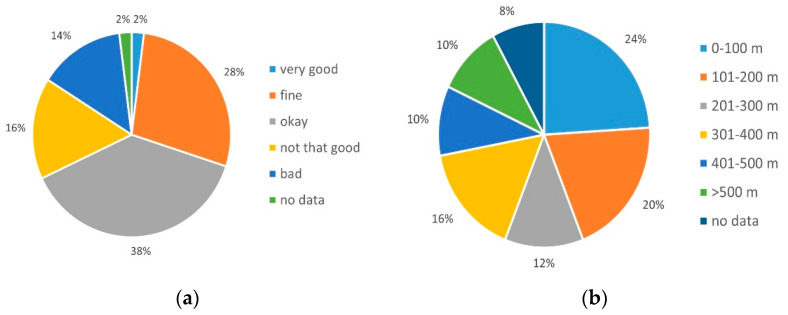
Health status: (**a**) current condition (self-assessment), (**b**) 6-minute-walk-distance.

**Figure 3 ijerph-19-03768-f003:**
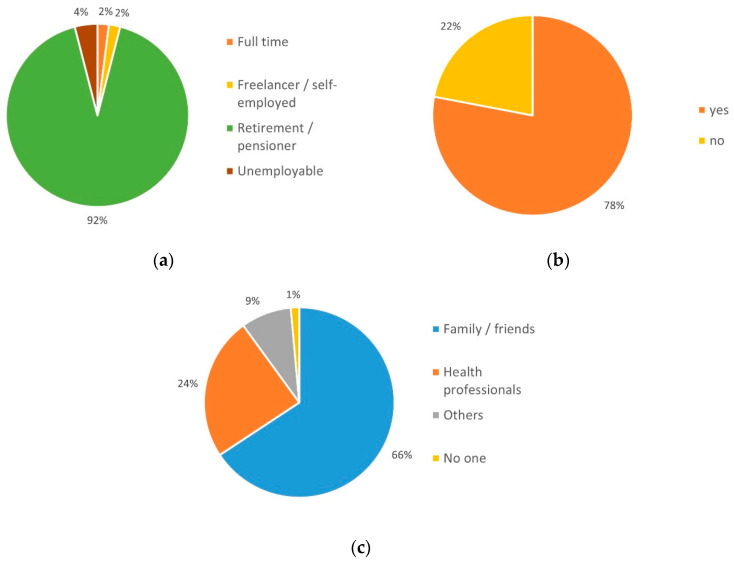
Situation at home: (**a**) job situation, (**b**) support at home, (**c**) primary care giver.

**Figure 4 ijerph-19-03768-f004:**
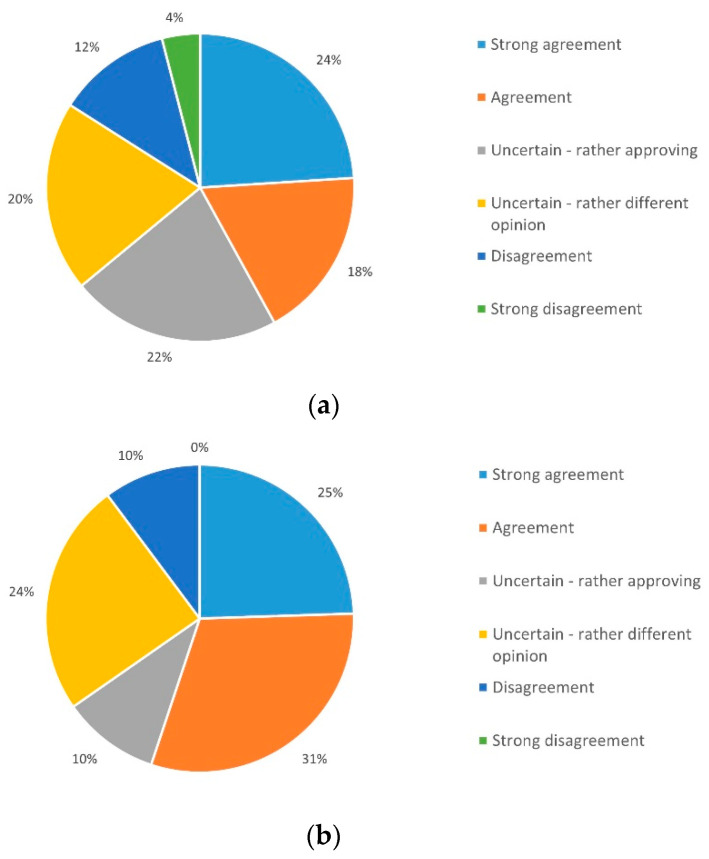
Expectation for improvements in terms of (**a**) individual health situation and (**b**) realization of ideas for improvements.

**Figure 5 ijerph-19-03768-f005:**
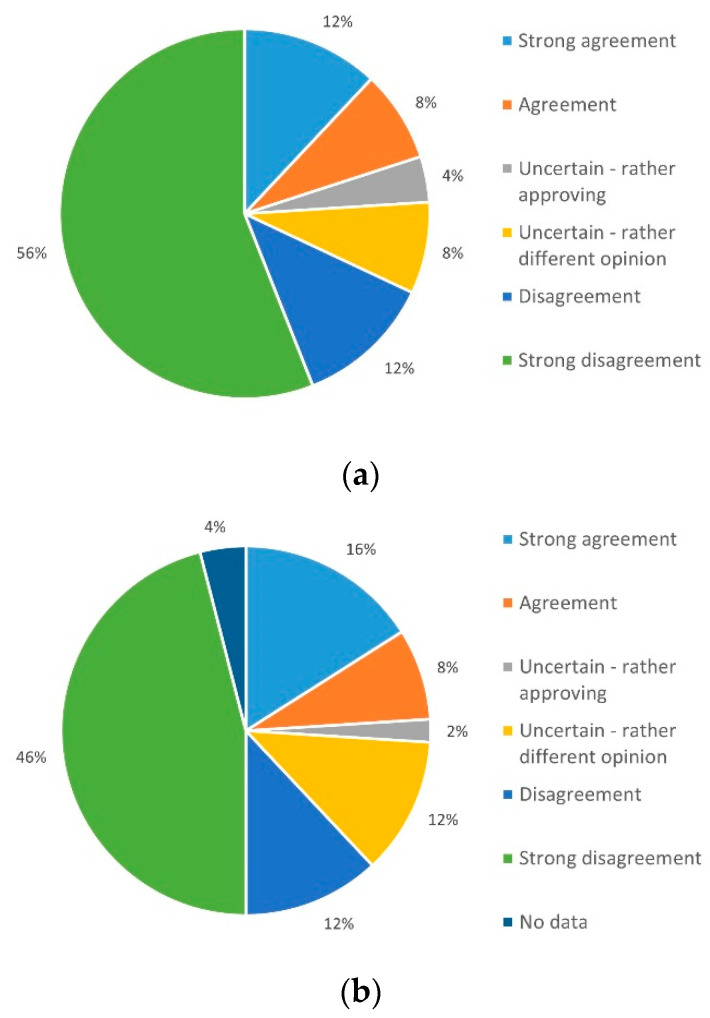
Expectation for problems of (**a**) privacy security issues and (**b**) daily use.

**Figure 6 ijerph-19-03768-f006:**
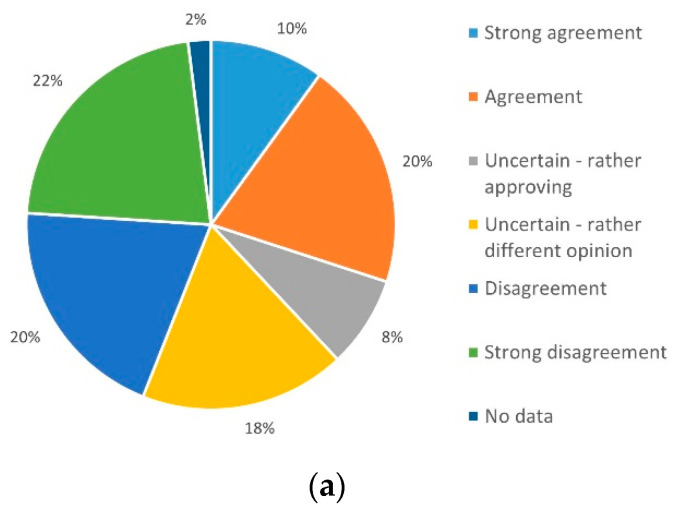
Concerns about (**a**) quality of care and (**b**) responsible staff.

**Figure 7 ijerph-19-03768-f007:**
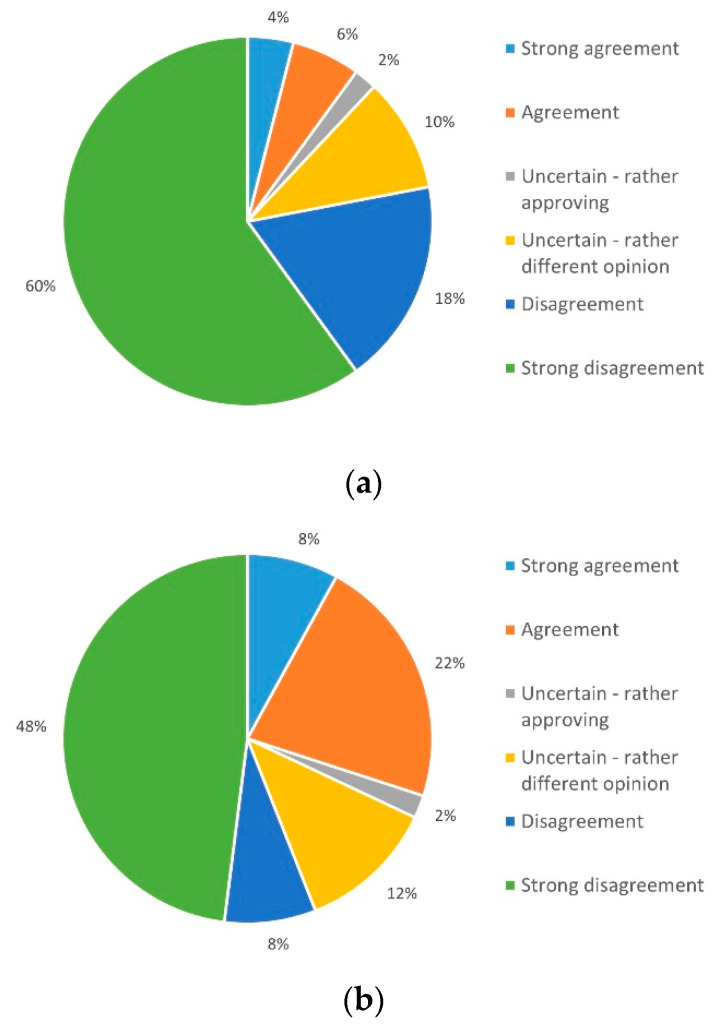
Expectations that (**a**) telemedicine affects individual health, (**b**) regular visits can be replaced, and (**c**) personal contacts will become obsolete.

**Table 1 ijerph-19-03768-t001:** Hospital contacts (average values).

Hospital Contacts		
Days of inpatient stay in 2017 Days of inpatient stay in 2018	19.48	(max 70)
	11.42	(max 60)
Visits in outpatient ward/12 months	6.81	

**Table 2 ijerph-19-03768-t002:** Efforts necessary to reach the clinic.

Efforts to Reach the Clinic		
Distance residence-to-clinic	49.63 km	(max 260 km)
Time to reach the clinic	<30 min: 48%	>30 min: 52%
Difficult to get to the clinic	No: 48%	Yes/partially: 52%

## Data Availability

The data presented in this study are available within the article.

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
