# Peer review of "VAD Patients’ Perception of Potential Telemedicine Support"

_ijerph, 2022, doi:10.3390/ijerph19073768_

Round 1

Reviewer 1 Report

It is interesting study, but there are some questions to be clarified.

 Is this study accepted in the local ethics committee? Please give accepted number in the method.

 It is possible that the answer would vary depending on the background of the patients. Please provide detailed demographics of the patient cohort. Please compare the difference of the answers between patients with long or short history after transplantation. I assume that patients with longer history or more stable situation would ask telemedicine than patients with not.

 When evaluating the possibility of telemedicine, it is important to note about the medical welfare support from the government. I assume that patients available for comprehensive support would be acceptable for telemedicine. Please note whether there are differences about services which patients obtained. Please discuss the relationship between situation of welfare service and the acceptance of telemedicine.

Reviewer 2 Report

My major comments concern the methods and analysis:

  1. The authors state that patients were selected randomly. As far as I can tell, this refers to which patients were asked to participate. But there is no information regarding how many patients agreed to or declined to participate. The participating patients may differ in important ways from those who declined. They may have stronger opinions about VAD. There is no information that allows us to assess this.
  2. Related to the above point: the authors do not assess whether the resulting sample is representative of the VAD patient population.
  3. The selection of patients is limited to those with VAD, but it could also be important to understand the reasons why some patients declined VAD if it was an option for them. This should at least be explored as a limitation.
  4. 50 is quite a small sample. Furthermore, there are no inferential statistics. As such, we really can't conclude whether certain reported percentages are greater than others. For example: the authors write, "In the second part the survey showed large differences among patients with regard to the number of their in-hospital stays in the year 2017 and 2018," but I highly doubt these percentages are statistically significant different from one another.
  5. I could not find a record of the statements that patients were presented with. This leads to two problems. First, it is impossible to assess whether the statements were worded comprehensibly or might have biased respondents in some way. Second, I cannot clearly interpret what the results mean. For example, the authors write, "42% (n=21) of respondents had positive expectations associated 189 with the use of telemedicine": Were respondents asked, "do you have positive expectations associated with the use of telemedicine?" or is this the author's summary of what the respondents were in fact asked?
  6. Related to the point above: Some of the findings include words like "expectations" which imply that patients were asked about their prospective opinions/perceptions. But the study is described as retrospective. Again, it is necessary to see exactly what participants were asked. It is also difficult to understand some of the findings. For example, I did not know what the following means: "expectations regarding a better exchange of experience with the clinic". This may simply be an issue of translation.
  7. I do not know what the percentages in the Results section represent. Participants answered on a Likert scale. Presumably the authors aggregated responses (such as strongly agree and agree) to arrive at these percentages, but this is not described. Furthermore, when I try to arrive at these percentages myself by adding percentages in the pie charts, I am finding inaccuracies. There are also percentages reported in the Discussion, but I am unclear where these come from.
  8. The authors say patient consent was waived due to anonymity, but this is not a sufficient reason to waive patient consent. 

In the spirit of constructive criticism, I suggest the authors consider doing a qualitative analysis of their findings (they said there were some open-ended responses). The sample size is small enough to represent a useful pilot study to uncover important issues for more rigorous, quantitative assessment. The authors might also consider delving deeper into differences among patients in their sample. For example, do patients who live further from care have more positive opinions of telemedicine (qualitatively) than those who live near to care?

Reviewer 3 Report

In the Abstract please explain what VAD is. Please improve the shape of the pie charts, as some of them are oval shaped. Also please try to put the caption of every figure just under the figure and not in the next page, as well as you should enlarge the bullets explaining the colours in pie charts. In paragraph "3.3.3 Privacy" please explain more the pie charts.

Reviewer 4 Report

Dear Authors,

You need considerable revision to improve your manuscript. 

  1. Please provide the scope and overview of the study in more detail (in the introduction section)
  2. The research design is ok but poor. Is it possible to analyze the data with a industry standard statistical software (Amos/Mplus/ Stata)? 
  3. Add practical significance/ implication
  4. Add managerial implication
  5. Add theoretical implication

Round 2

Reviewer 1 Report

Authors answered my questions appropriately.

Author Response

Thank you for your support in the review of our paper; You have helped us a lot with your advice to achieve a better result!

Kind regards

Reviewer 2 Report

see attached

Reviewer 4 Report

The revised version is better. Thank you. 

Author Response

Thank you for your support in the review of our paper; You have helped us a lot with your advice to achieve a better result!

The professional language editing will be done promptly.

Kind regards

This manuscript is a resubmission of an earlier submission. The following is a list of the peer review reports and author responses from that submission.